# MaskRNN: Instance Level Video Object Segmentation

**Yuan-Ting Hu**
UIUC
ythu2@illinois.edu

**Jia-Bin Huang**
Virginia Tech
jbhuang@vt.edu

**Alexander G. Schwing**
UIUC
aschwing@illinois.edu

## Abstract

Instance level video object segmentation is an important technique for video editing and compression. To capture the temporal coherence, in this paper, we develop MaskRNN, a recurrent neural net approach which fuses in each frame the output of two deep nets for each object instance — a binary segmentation net providing a mask and a localization net providing a bounding box. Due to the recurrent component and the localization component, our method is able to take advantage of long-term temporal structures of the video data as well as rejecting outliers. We validate the proposed algorithm on three challenging benchmark datasets, the DAVIS-2016 dataset, the DAVIS-2017 dataset, and the Segtrack v2 dataset, achieving state-of-the-art performance on all of them.

## 1 Introduction

Instance level video object segmentation of complex scenes is a challenging problem with applications in areas such as object identification, video editing, and video compression. With the recent release of the DAVIS dataset [39], the task of segmenting multiple object instances from videos has gained considerable attention. However, just like for classical foreground-background segmentation, deforming shapes, fast movements, and multiple objects occluding each other pose significant challenges to instance level video object segmentation.

Classical techniques [5, 10, 11, 17, 21, 41, 20, 44, 49] for video object segmentation often rely on geometry and assume rigid scenes. Since these assumptions are often violated in practice, visually apparent artifacts are commonly observed. To temporally and spatially smooth object mask estimates, graphical model based techniques [22, 2, 14, 45, 47, 46] have been proposed in the past. While graphical models enable an effective label propagation across the entire video sequences, they often tend to be sensitive to parameters.

Recently, deep learning based approaches [7, 26, 23, 6, 25] have been applied to video object segmentation. Early work in this direction predicts the segmentation mask frame by frame [7]. Later, prediction of the current frame incorpoerates additional cues from the preceding frame using optical flow [23, 26, 25], semantic segmentations [6], or mask propagation [26, 25]. Importantly, all these methods only address the foreground-background segmentation of a single object and are not directly applicable to instance level segmentation of multiple objects in videos.

In contrast to the aforementioned methods, in this paper, we develop MaskRNN, a framework that deals with *instance level segmentation* of multiple objects in videos. We use a bottom-up approach where we first track and segment individual objects before merging the results. To capture the temporal structure, our approach employs a recurrent neural net while the segmentation of individual objects is based on predictions of binary segmentation masks confined to a predicted bounding box.

We evaluate our approach on the DAVIS-2016 dataset [37], the DAVIS-2017 dataset [39], and the Segtrack v2 dataset [30]. On all three we observe state-of-the-art performance.

Table 1: Comparisons with the state-of-the-art deep learning based video object segmentation algorithms.

| Method | OSVOS [7] | MaskProp [26] | FusionSeg [23] | LucidTracker [25] | SemanticProp [6] | Ours |
|---|---|---|---|---|---|---|
| Using flow | No | Yes | Yes | Yes | No | Yes |
| Temporal information | No | Short-term | Short-term | Short-term | No | Long-term (RNN) |
| Location prior | No | Previous mask | No | Previous mask | No | Previous mask+Bounding box |
| Semantic prior | No | No | No | No | Yes | No |
| Post-processing | Boundary snapping | denseCRF | No | denseCRF | No | No |
| Finetuning on the 1st frame | Yes | Yes | No | Yes | Yes | Yes |

## 2    Related Work

Video object segmentation has been studied extensively in recent years [45, 30, 34, 40, 29, 28, 36, 48, 16, 46, 37, 23, 6, 25]. In the following, we group the literature into two categories: (1) graph-based approaches and (2) deep learning methods.

**Video object segmentation via spatio-temporal graphs:** Methods in this category construct a three-dimensional spatio-temporal graph [45, 30, 16, 28] to model the inter- and the intra-frame relationship of pixels or superpixels in a video. Evidence about a pixels assignment to the foreground or background is then propagated along this spatio-temporal graph, to determine which pixels are to be labeled as foreground and which pixel corresponds to the background of the observed scene. Graph-based approaches are able to accept different degrees of human supervision. For example, interactive video object segmentation approaches allow users to annotate the foreground segments in several key frames to generate accurate results by propagating the user-specified masks to the entire video [40, 13, 34, 31, 22]. Semi-supervised video object segmentation techniques [4, 16, 45, 22, 46, 33] require only one mask for the first frame of the video. Also, there are unsupervised methods [9, 28, 50, 36, 35, 12, 48] that do not require manual annotation. Since constructing and exploring the 3D spatio-temporal graphs is computationally expensive, the graph-based methods are typically slow, and the running time of the graph-based video object segmentation is often far from real time.

**Video object segmentation via deep learning:** With the success of deep nets on semantic segmentation [32, 42], deep learning based approaches for video object segmentation [7, 26, 23, 6, 25] have been intensively studied recently and often yield state-of-the-art performance, outperforming graph-based methods. Generally, the employed deep nets are pre-trained on object segmentation datasets. In the semi-supervised setting where the ground truth mask of the first frame of a video is given, the network parameters are then finetuned on the given ground truth of the first frame of a particular video, to improve the results and the specificity of the network. Additionally, contour cues [7] and semantic segmentation information [7] can be incorporated into the framework. Besides those cues, optical flow between adjacent frames is another important key information for video data. Several methods [26, 23, 25] utilize the magnitude of the optical flow between adjacent frames. However, these methods do not explicitly model the location prior, which is important for object tracking. In addition, these methods focus on separating foreground from background and do not consider instance level segmentation of multiple objects in a video sequence.

In Tab. 1, we provide a feature-by-feature comparison of our video object segmentation technique with representative state-of-the-art approaches. We note that the developed method is the only one that takes long-term temporal information into account via back-propagation through time using a recurrent neural net. In addition, the discussed method is the only one that estimates the bounding boxes in addition to the segmentation masks, allowing us to incorporate a location prior of the tracked object.

## 3    Instance Level Video Object Segmentation

Next, we present MaskRNN, a joint multi-object video segmentation technique, which performs instance level object segmentation by combining binary segmentation with effective object tracking via bounding boxes. To benefit from temporal dependencies, we employ a recurrent neural net component to connect prediction over time in a unifying framework. In the following, we first provide a general outline of the developed approach illustrated in Fig. 1 and detail the individual components subsequently.

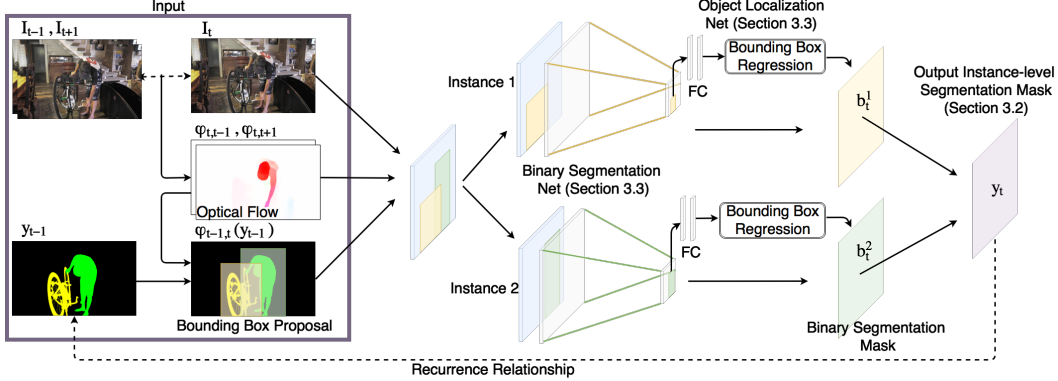

Figure 1: An illustration for the proposed algorithm. We show an example video with 2 objects (left). Our method predicts the binary segmentation for each object using 2 deep nets (Section 3.3), one for each object, which perform binary segmentation and object localization. The output instance-level segmentation mask is obtained by combining the binary segmentation masks (Section 3.2).

## 3.1 Overview

We consider a video sequence $I = \{I_1, I_2, ..., I_T\}$ which consists of $T$ frames $I_t$, $t \in \{1, ..., T\}$. Throughout, we assume the ground truth segmentation mask of the $N$ object instances of interest to be given for the first frame $I_1$. We refer to the ground truth segmentation mask of the first frame via $y_1^* \in \{0, 1, ..., N\}^{H \times W}$, where $N$ is the number of object instances, and $H$ and $W$ are the height and width of the video frames. In multi-instance video object segmentation, the goal is to predict $y_2, ..., y_T \in \{0, ..., N\}^{H \times W}$, which are the segmentation masks corresponding to frames $I_2$ to $I_T$.

The proposed method is outlined in Fig. 1. Motivated by the time-dependence of the frames in the video sequence we formulate the task of instance level semantic video segmentation as a recurrent neural net, where the prediction of the previous frame influences prediction of the current frame. Beyond the prediction $y_{t-1}$ for the previous frame $t-1$, our approach also takes into account both the previous and the current frames, *i.e.*, $I_{t-1}$ and $I_t$. We compute the optical flow from the two images. We then use the predicted optical flow (i) as input feature to the neural nets and (ii) to warp the previous prediction to roughly align with the current frame.

The warped prediction, the optical flow itself, and the appearance of the current frame are then used as input for $N$ deep nets, one for each of the $N$ objects. Each of the deep nets consists of two parts, a binary segmentation net which predicts a segmentation mask, and an object localization net which performs bounding box regression. The latter is used to alleviate outliers. Both, bounding box regression and segmentation map are merged into a binary segmentation mask $b_t^i \in [0, 1]^{H \times W}$ denoting the foreground-background probability maps for each of the $N$ object instances $i \in \{1, ..., N\}$. The binary semantic segmentations for all $N$ objects are subsequently merged using an arg max operation. The prediction for the current frame, *i.e.*, $y_t$, is computed via thresholding. Note that we combine the binary predictions only at test time.

In the following, we first describe our fusion operation in detail, before discussing the deep net performing binary segmentation and object localization.

## 3.2 Multiple instance level segmentation

Predicting the segmentation mask $y_t$ for the $t$-th frame, can be viewed as a multi-class prediction problem, *i.e.*, assigning to every pixel in the video a label, indicating whether the pixel $p$ represents an object instance ($y_t^p = \{1, ..., N\}$) or whether the pixel is considered background ($y_t^p = 0$). Following a recent technique for instance level image segmentation [18], we cast this multi-class prediction problem into multiple binary segmentations, one per object instance.

Assume availability of binary segmentation masks $b_t^i \in [0, 1]^{H \times W}$ which provide for each object instance $i \in \{1, ..., N\}$ the probability that a pixel should be considered foreground or background. To combine the binary segmentations $b_t^i$ into one final prediction $y_t$ such that every pixel is assigned to only one object label, is achieved by assigning the class with the largest probability for every pixel.

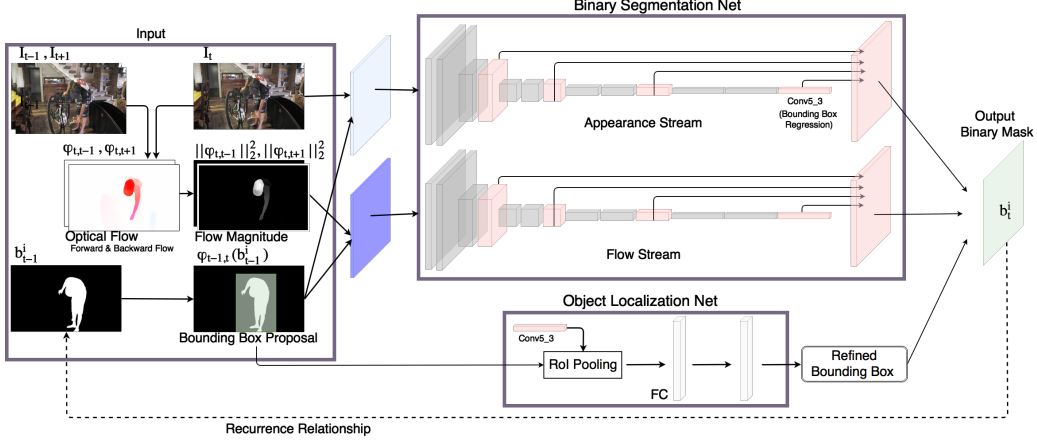

Figure 2: An illustration of the binary object segmentation network and the object localization network as described in Section 3.3. The binary segmentation network is a two-stream network including appearance stream and flow stream. The inputs of the appearance stream are the current frame $I_t$ and $\phi_t(b_{t-1}^i)$. The inputs of the flow stream are the flow magnitude and the warped mask, $\phi_t(b_{t-1}^i)$. The object localization net refines the bounding box proposal to estimate the location prior. To compute the binary segmentation mask $b_t^i$, the output of appearance stream and the flow stream are linearly combined and the responses outside the refined bounding box are discarded.

To be more specific, we assign class label $i \in \{1, \ldots, N\}$ to the pixel if the probability for class $i$ at the pixel (indicated by $b_t^i$) is largest among the $N$ probability maps for the $N$ object instances. Note that this operation is similar to a pooling operation, and permits back-propagation.

## 3.3 Binary Segmentation

To obtain the binary segmentations $b_t^i \in [0, 1]^{H \times W}$ employed in the fusion step, $N$ deep nets are used, one for each of the $N$ considered object instances. One of the $N$ deep nets is illustrated in Fig. 2. It consists of two components, the binary segmentation net and the object localization net, which are discussed in greater detail in the following.

**Binary Segmentation Net:** The objective for each of the binary segmentation nets is to predict the foreground-background mask $b_t^i \in [0, 1]^{H \times W}$ for its corresponding object instance $i \in \{1, \ldots, N\}$. To achieve this task, the binary segmentation net is split into two streams, *i.e.*, the appearance stream and the flow stream. The input of the appearance stream is the concatenation of the current frame $I_t$ and the warped prediction of the previous frame $y_{t-1}$, denoted as $\phi_{t-1,t}(y_{t-1})$. The warping function $\phi_{t-1,t}(.)$ transforms the input based on the optical flow field from frame $I_{t-1}$ to frame $I_t$. The input of the flow stream is the concatenation of the magnitude of the flow field from $I_t$ to $I_{t-1}$ and $I_t$ to $I_{t+1}$ and, again, the warped prediction of the previous frame $\phi_{t-1,t}(y_{t-1})$. The architecture of both streams is identical and follows the subsequent description.

The network architecture is inspired by [7] where the bottom of the network follows the structure of the VGG-16 network [43]. The intermediate representations of the VGG-16 network, right before the max-pooling layers and after the ReLU layers, are extracted, upsampled by bilinear interpolation and linearly combined to form a single channel feature representation which has the same size as the input image. By linearly combining the two representations, one from the appearance stream and the other one from the flow stream, and by taking the sigmoid function on the combined single channel feature response, we obtain a probability map which indicates the probability $b_t^i \in [0, 1]^{H \times W}$ of a pixel in the $t$-th frame being foreground, *i.e.*, corresponding to the $i$-th object. The network architecture of the appearance stream is shown in Fig. 2 (right panel). During training, we use the weighted binary cross entropy loss as suggested in [7].

Note that all the operations in our network are differentiable. Hence, we can train the developed network end-to-end via back-propagation through time.

Table 2: Contribution of different components of our algorithm evaluated on DAVIS-2016 and DAVIS-2017 dataset. The performance is in term of IoU (%).

| Component | | Enable (✓) / Disable | | | | | |
|---|---|---|---|---|---|---|---|
| Segmentation net | AStream | ✓ | ✓ | ✓ | ✓ | ✓ | ✓ |
| | FStream | | ✓ | ✓ | ✓ | ✓ | ✓ |
| | Warp mask | | | ✓ | ✓ | ✓ | ✓ |
| Localization net | Train | | | | ✓ | ✓ | ✓ |
| | Apply | | | | | ✓ | ✓ |
| RNN | | | | | | | ✓ |
| DAVIS-2016 IoU(%), w/o Finetuning | | 54.17 | 55.87 | 56.88 | 52.29 | 53.90 | 56.32 |
| DAVIS-2016 IoU(%), w/ Finetuning | | 76.63 | 79.77 | 79.92 | 78.43 | 80.10 | **80.38** |
| DAVIS-2017 IoU(%), w/o Finetuning | | 41.29 | 43.33 | 44.52 | 38.95 | 41.57 | 45.53 |
| DAVIS-2017 IoU(%), w/ Finetuning | | 58.66 | 59.46 | 59.71 | 56.12 | 60.41 | **60.51** |

**Object Localization Net:** Usage of an object localization net is inspired by tracking approaches which regularize the prediction by assuming that the object is less likely to move drastically between temporally adjacent frames. The object localization network computes the location for the $i$-th object in the current frame via bounding box regression. First, we find the bounding box proposal on the warped mask $\phi_t(b_{t-1}^i)$. Similarly to the bounding box regression in Fast-RCNN [15], with the bounding box proposal as the region of interest, we use the conv5_3 feature in the appearance stream of the segmentation net to perform RoI-pooling, followed by two fully connected layers. Their output is used to regress the bounding box position. We refer the reader to [15] for more details on bounding box regression.

Given the bounding box, a pixel is classified as foreground if it is predicted as foreground by the segmentation net and if it is inside a bounding box which is enlarged by a factor of 1.25 compared to the predicted of the localization net. The estimated bounding box is then used to restrict the segmentation to avoid outliers which are far away from the object.

### 3.4 Training and Finetuning

Our framework outlined in the preceding sections and illustrated in Fig. 1 can be trained end-to-end via back-propagation through time given a training sequence. Note that back-propagation through time is used because of the recurrence relation that connects multiple frames of the video sequence. To further improve the predictive performance, we follow the protocol [39] for the semi-supervised setting of video object segmentation and finetune our networks using the ground truth segmentation mask provided for the first frame. Specifically, we further optimize the binary segmentation net and localization net based on the given ground truth. Note that it is not possible to adjust the entire architecture since only a single ground truth frame is provided in the supervised setting.

## 4 Implementation Details

In the following, we describe the implementation details of our approach, as well as the training data. We also provide details about the offline training and online training in our experimental setup.

**Training data:** We use the training set of the DAVIS dataset to pre-train the appearance network for general-purpose object segmentation. The DAVIS-2016 dataset [37] contains 30 training videos and 20 testing videos and the DAVIS-2017 dataset [39] consists of 60 training videos and 30 testing videos. Note that the annotation of the DAVIS-2016 dataset contains only one single object per video. For a fair evaluation on the DAVIS-2016 and DAVIS-2017 datasets, the object segmentation net and localization nets are trained on the training set of each dataset separately. During testing, the network is further finetuned online on the given ground-truth of the first frame since we assume the ground truth segmentation mask of the first frame, $i.e.$, $y_1^*$, to be available.

**Offline training:** During offline training, we first optimize the networks on static images. We found it useful to randomly perturb the ground-truth segmentation mask $y_{t-1}^*$ locally, to simulate the imperfect prediction of the last frame. The random perturbation includes dilation, deformation, resizing, rotation and translation. After having trained both the binary segmentation net and the object localization net on single frames, we further optimize the segmentation net by taking long-term

Table 3: The quantitative evaluation on the validation set of DAVIS dataset [37]. The evaluation matrics are the IoU measurement $\mathscr{J}$, boundary precision $\mathscr{F}$, and time stability $\mathscr{T}$. Following [37], we also report the recall and the decay of performance over time for $\mathscr{J}$ and $\mathscr{F}$ measurements.

| | | Semi-supervised | | | | | | | | | | |
| | | OSVOS [7] | MSK [26] | VPN [24] | OFL [46] | BVS [33] | FCP [38] | JMP [13] | HVS [16] | SEA [1] | TSP [8] | **OURS** |
|---|---|---|---|---|---|---|---|---|---|---|---|---|
| | Mean $\mathscr{M}\uparrow$ | 79.8 | 79.7 | 70.2 | 68.0 | 60.0 | 58.4 | 57.0 | 54.6 | 50.4 | 31.9 | **80.4** |
| $\mathscr{J}$ | Recall $\mathscr{O}\uparrow$ | 93.6 | 93.1 | 82.3 | 75.6 | 66.9 | 71.5 | 62.6 | 61.4 | 53.1 | 30.0 | **96.0** |
| | Decay $\mathscr{D}\downarrow$ | 14.9 | 8.9 | 12.4 | 26.4 | 28.9 | **-2.0** | 39.4 | 23.6 | 36.4 | 38.1 | 4.4 |
| | Mean $\mathscr{M}\uparrow$ | 80.6 | 75.4 | 65.5 | 63.4 | 58.8 | 49.2 | 53.1 | 52.9 | 48.0 | 29.7 | **82.3** |
| $\mathscr{F}$ | Recall $\mathscr{O}\uparrow$ | 92.6 | 87.1 | 69.0 | 70.4 | 67.9 | 49.5 | 54.2 | 61.0 | 46.3 | 23.0 | **93.2** |
| | Decay $\mathscr{D}\downarrow$ | 15.0 | 9.0 | 14.4 | 27.2 | 21.3 | **-1.1** | 38.4 | 22.7 | 34.5 | 35.7 | 8.8 |
| $\mathscr{T}$ | Mean $\mathscr{M}\downarrow$ | 37.8 | 21.8 | 32.4 | 22.2 | 34.7 | 30.6 | 15.9 | 36.0 | **15.4** | 41.7 | 19.0 |

information into account, *i.e.*, training using the recurrence relation. We consider 7 frames at a time due to the memory limitation imposed by the GPU.

During offline training all networks are optimized for 10 epochs using the Adam solver [27] and the learning rate is gradually decayed during training, starting from $10^{-5}$. Note that we use the pre-trained flowNet2.0 [19] for optical flow computation. During training, we apply data augmentation with randomly resizing, rotating, cropping, and left-right flipping the images and masks.

**Online finetuning:** In the semi-supervised setting of video object segmentation, the ground-truth segmentation mask of the first frame is available. The object segmentation net and the localization net are further finetuned on the first frame of the testing video sequence. We set the learning rate to $10^{-5}$. We train the network for 200 iterations, and the learning rate is gradually decayed over time. To enrich the variation of the training data, for online finetuning the same data augmentation techniques are applied as in offline training, namely randomly resizing, rotating, cropping and flipping the images. Note that the RNN is *not* employed during online finetuning since only a single frame of training data is available.

## 5   Experimental Results

Next, we first describe the evaluation metrics before we present an ablation study of our approach, quantitative results, and qualitative results.

### 5.1   Evaluation Metrics

**Intersection over union:** We use the common mean intersection over union (IoU) metric which calculates the average across all frames of the dataset. The IoU metric is particularly challenging for small sized foreground objects.

**Contour accuracy [37]:** Besides an accurate object overlap measured by IoU, we are also interested in an accurate delineation of the foreground objects. To assess the delineation quality of our approach, we measure the precision, $P$, and the recall $R$ of the two sets of points on the contours of the ground truth segment and the output segment via a bipartite graph matching. The contour accuracy is calculated as $\frac{2PR}{P+R}$.

**Temporal stability [37]:** The temporal stability estimates the degree of deformation needed to transform the segmentation masks from one frame to the next. The temporal stability is measured by the dissimilarity of the shape context descriptors [3] which describe the points on the contours of the segmentation between the two adjacent frames.

### 5.2   Ablation study

We validate the contributions of the components in our method by presenting an ablation study summarized in Tab. 2 on two datasets, DAVIS-2016 and DAVIS-2017. We mark the enabled components using the '✓' symbol. We analyze the contribution of the binary segmentation net

Table 4: The quantitative evaluation on DAVIS-2017 dataset [39] and SegTrack v2 dataset [30].

|  | DAVIS-2017 | | | SegTrack v2 | | | |
|---|---|---|---|---|---|---|---|
|  | OSVOS [7] | OFL [46] | **OURS** | OSVOS [7] | MSK [26] | OFL [46] | **OURS** |
| IoU(%) | 52.1 | 54.9 | **60.5** | 61.9 | 67.4 | 67.5 | **72.1** |

including the appearance stream ('AStream'), the flow stream ('FStream') and whether to warp the input mask, $y_{t-1}$, based on the optical flow field ('Warp mask'). In addition, we analyze the effects of the object localization net. Specifically, we assess the occurring performance changes of two configurations: (i) by only adding the bounding box regression loss into the objective function ('Train'), *i.e.*, both the segmentation net and the object localization net are trained but only the segmentation net is deployed; (ii) by training and applying the object localization net ('Apply'). The contribution of the recurrent training ('RNN') is also illustrated. The performances with and without online finetuning as described in Section 4 are shown for each dataset as well.

In Tab. 2, we generally observe that online finetuning is important as the network is adjusted to the specific object appearance in the current video.

For the segmentation net, the combination of the appearance stream and the flow stream performs better than using only the appearance stream. This is due to the fact that the optical flow magnitude provided in the flow stream provides complementary information by encoding motion boundaries, which helps to discover moving objects in the cluttered background. The performance can be further improved by using the optical flow to warp the mask so that the input to both streams of the segmentation net also takes the motion into account.

For the localization net, we first show that adding the bounding box regression loss decreases the performance of the segmentation net (adding 'Train' configuration). However, by applying the bounding box to restrict the segmentation mask improves the results beyond the performance achieved by only applying the segmentation net.

Training the network using the recurrence relationship further improves the results as the network produces more consistent segmentation masks over time.

## 5.3 Quantitative evaluation

We compare the performance of our approach to several baselines on two tasks: foreground-background video object segmentation and multiple instance-level video object segmentation. More specifically, we use DAVIS-2016 [37] for evaluating foreground-background segmentation, and DAVIS-2017 [39] and Segtrack v2 [30] datasets for evaluating multiple instance-level segmentation. The three datasets serve as a good testbed as they contain challenging variations, such as drastic appearance changes, fast motion, and occlusion. We compare the performance of our approach to several state-of-the-art benchmarks. We assess performance on the validation set when using the DAVIS datasets and we use the whole dataset for Segtrack v2 as no split into train and validation sets is available. The results on DAVIS-2016 are summarized in Tab. 3, where we report the IoU, the contour accuracy, and the time stability metrics following [37]. The results on DAVIS-2017 and SegTrack v2 are summarized in Tab. 4.

**Foreground-background video object segmentation:** We use the DAVIS-2016 dataset to evaluate the performance of foreground-background video object segmentation. The DAVIS-2016 dataset contains 30 training videos and 20 validation videos. The network is first trained on the 30 training videos and finetuned on the first frame of the 20 validation videos, respectively during testing. The performance evaluation is reported in Tab. 3. We outperform the other state-of-the-art semi-supervised methods by 0.6%. Note that OSVOS [7], MSK [26], VPN [24] are also deep learning approach. In contrast to our approach, these methods don't employ the location prior.

**Instance-level video object segmentation:** We use the DAVIS-2017 and the Segtrack v2 datasets to evaluate the performance of instance-level video object segmentation. The DAVIS-2017 dataset contains 60 training videos and 30 validation videos. The Segtrack v2 dataset contains 14 videos. There are 2.27 objects per video on average in the DAVIS-2017 dataset and 1.74 in the Segtrack v2 dataset. Again, as for DAVIS-2016, the network is trained on the training set and then finetuned using the groundtruth of the given first frame. Since the Segtrack v2 dataset does not provide a training set, we use the DAVIS-2017 training set to optimize and finetune the deep nets. The performance evaluation is reported in Tab. 4. We outperform other state-of-the-art semi-supervised methods by 5.6% and 4.6% on DAVIS-2017 and Segtrack v2, respectively.

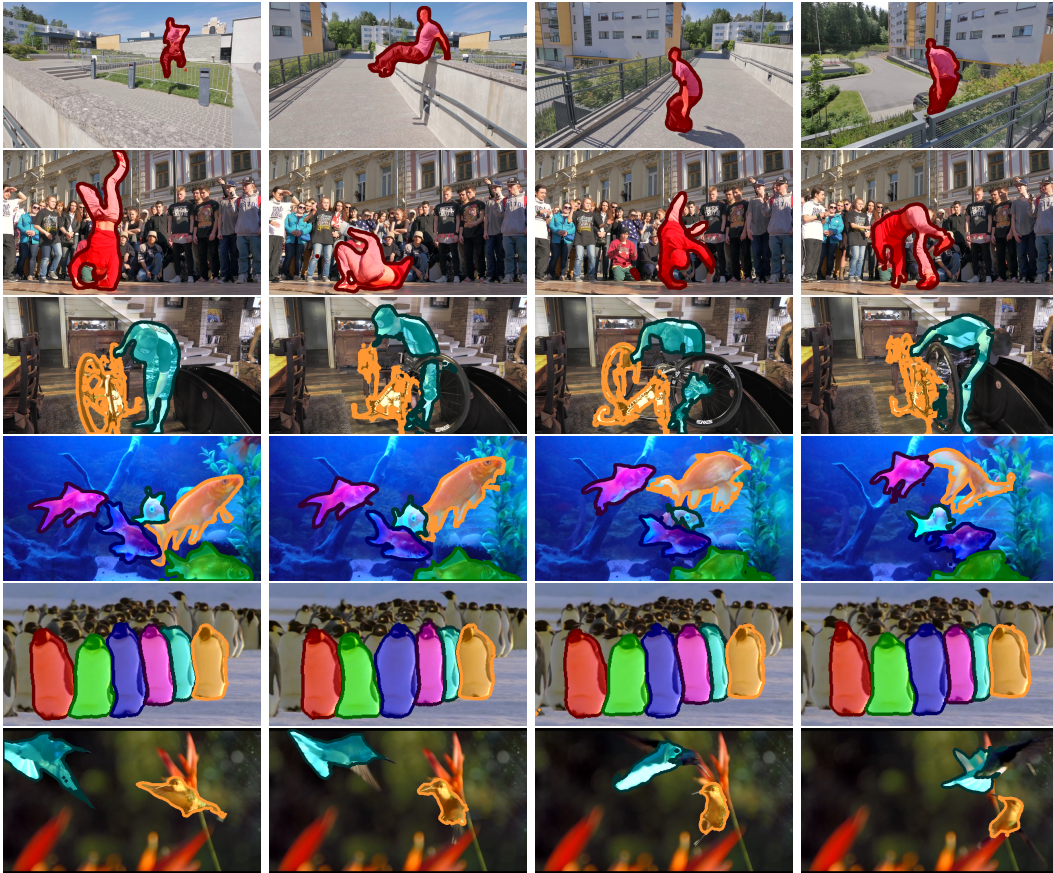

Figure 3: Visual results of our approach on DAVIS-2016 (1st and 2nd row), DAVIS-2017 (3rd and 4th row) and Segtrack v2 dataset (5th and 6th row).

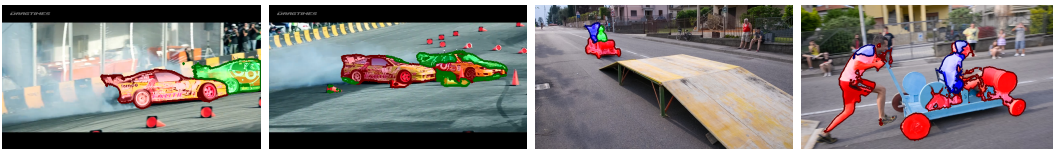

Figure 4: Failure cases of our approach. The 1st and the 3rd column shows the results of the beginning frames. Our method fails to track the object instances as shown in the 2nd and 4th column.

## 5.4 Qualitative evaluation

We visualize some of the qualitative results of our approach in Fig. 3 and Fig. 4. In Fig. 3, we show some successful cases of our algorithm on the DAVIS and Segtrack datasets. We observe that the proposed method accurately keeps track of the foreground objects even with complex motion and cluttered background. We also observe accurate instance level segmentation of multiple objects which occlude each other. In Fig. 4, we visualize two failure cases of our approach. Reasons for failures are the similar appearance of instances of interest as can be observed for the leftmost two figures. Another reason for failure is large variations in scale and viewpoint as shown for the two figures on the right of Fig. 4.

## 6 Conclusion

We proposed MaskRNN, a recurrent neural net based approach for instance-level video object segmentation. Due to the recurrent component and the combination of segmentation and localization nets, our approach takes advantage of the long-term temporal information and the location prior to improve the results.

**Acknowledgments:** This material is based upon work supported in part by the National Science Foundation under Grant No. 1718221. We thank NVIDIA for providing the GPUs used in this research.

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
