[Reviews · NeurIPS 2017]

Reviewer 1



This manuscript introduces an approach for Instance-level video object segmentation based on a recurrent neural net, which can capture the temporal coherence and fuses in each frame the outputs of two deep nets for each object instance: a binary segmentation net providing a mask and a localization net providing a bounding box. The experimental results demonstrate the advantage of the approach. Overall, the approach looks reasonable and the text is well-written. However, there are some issues in the manuscript and it needs a careful revision to remove all such issues. To name a few: - Figure 1 and Figure 2 look a bit redundant and can be fused in some way. - In line 127-129: it is stated that the input of the flow stream is the concatenation of the magnitude of the flow field from I_{t} to I_{t-1} and I_{t} to I_{t+1} and, again, the warped prediction of the previous frame \phi(t (y_{t-1}). However, this is not consistent with Figure 2. - The sentence in line 114-115 lacks some text.

Reviewer 2



Paper summary ---------------- This paper presets a recurrent neural network approach to instance segmentation in videos. The method is a combination of several techniques (RNNs, bounding box proposals, optical flow, binary segmentation), that, in combination, yield best performance on the current standard DAVIS dataset for this task. The techniques itself are known, the novelty lies in the combination into a system that performs well. A well done ablation study gives some insights in what works best for this problem (please include the effect of fine-tuning on first frame as well). Review summary ----------------- While not technically novel, the combined approach gives some insights in what works and why for this problem. This is likely of interest to researchers in this field. The paper is well written and experiments are well done, including a comprehensive ablation study. I have not seen this particular combination of models. In summary, I recommend acceptance of this submission, it is not a technical strong one, but presents an architecture that is among the best for video instance segmentation. I am not excited about this problem and have to admit that I am not fully aware of the literature on this problem. However, the submission is complete in the sense, that it achieves what it sets out to do and the combined system works well. Details / comments ------------------- - I recommend to include “fine-tuning on first frame” as a criterion for Table 1 and also for the ablation study. This technique is found to always improve performance but some methods do without. Since this severely effects runtime and the field should strive for instance independent mask propagation, methods that do without should be preferred. - What if the object is not moving and the optical flow is zero everywhere? - Please state the test time per frame / video sequence. typo: line 64 the the

Reviewer 3



Last year has seen a large progress in video object segmentation, triggered by the release of the DAVIS dataset. In particular, [Khoreva et al., CVPR'17] and [Caelles et al., CVPR'17] have shown that training a fully-convolutional network on the first frame of the video can greatly improve performance in the semi-supervised scenario, where the goal is to, given ground truth annotation in the first frame of a video, segment the annotated object in the remaining frames. In addition, Khoreva et al., demonstrated that feeding the prediction for the previous frame as an additional input to the network and utilising a parallel branch that operates on the optical flow magnitude further improves the performance. These approaches have indeed set the new state-of-the-art on DAVIS, but remained constrained to operating on a single frame and a single object at a time. In this work the authors propose to address these limitations as well as integrate some promising techniques from the recent object detection methods into a video object segmentation framework. Starting from the approach of Khoreva et al., which consists of two fully-convolutional networks pertained for mask refinement, taking rgb and flow magnitude as input respectively, as well as an object mask predicted in the previous frame, and fine-tuned on the first frame of the test video, they propose the following extensions: 1. The object mask from the previous frame is warped with optical flow to simplify mask refinement. 2. The model's prediction is "cleaned up" by taking a box proposal, refining it with a specialised branch of the network and suppressing all the segmentation predictions outside of the box. 3. A (presumably convolutional) RNN is attached in the end of the pipeline, which enforces temporal consistency on the predictions. 4. The model is extended to the instance-level scenarios, that is, it can segment multiple objects at once, assigning a distinct label to each of them. A complex system like this is, of course, challenging to present in an 8 page paper. The authors address this issue by simply skipping the description of some of the components. In particular, the RNN is not described at all and the details of the bounding box generation are omitted. Moreover, the "instance-level" results are obtained by simply running the model separately for each instance and merging the predictions on the frame level, which can just as well be done for any other method. In general, the paper is well written and the evaluation is complete, studying the effect of each extension in separation. The study shows, that in the scenario when the networks are fine-tuned on the first frame of the video, the effect of all the proposed extensions is negligible. This is in part to be expected, since, as was shown in the recent work, training a network on the first frame of a video serves as a very strong baseline in itself. When no fine-tuning on the test sequence is done though, the proposed extensions do improve the method's performance, but the final result in this setting on DAVIS'16 remains more than 13% bellow the rgb-only, no-test-finetuning model of Khoreva et al. Though their model is trained on a larger COCO dataset, this is hardly enough to explain such a large performance gap with a much simpler model. In the state-of-the-art comparison the full method outperforms the competitors on most of the measures on 3 datasets (DAVIS'16, DAVIS'17 and Seg-Track-v2). That said, the comparison with [Khoreva et al., CVPR'17] on DAVIS'17 is missing and the performance gap is not always significant relatively to the complexity of the proposed solution (0.9% on DAVIS'16). Especially, given the recent extension of Khoreva et al., (see reference [25] in the paper), which shows significantly better results with much simpler tools. Overall, the authors have done a good job by combining insights from several recent publications into a single framework but failed to achieve convincing results.